# GUD: Generation with Unified Diffusion

## Abstract

Diffusion generative models transform noise into data by inverting a process that progressively adds noise to data samples. Inspired by concepts from the renormalization group in physics, which analyzes systems across different scales, we revisit diffusion models by exploring three key design aspects: 1) the choice of representation in which the diffusion process operates (e.g. pixel-, PCA-, Fourier-, or wavelet-basis), 2) the prior distribution that data is transformed into during diffusion (e.g. Gaussian with covariance $\Sigma$), and most importantly 3) the scheduling of noise levels applied separately to different parts of the data, captured by a component-wise noise schedule. Incorporating the flexibility in these choices, we develop a unified framework for diffusion generative models with greatly enhanced design freedom. In particular, we introduce soft-conditioning models that smoothly interpolate between standard diffusion models and autoregressive models (in any basis), conceptually bridging these two approaches. Our framework opens up a wide design space which may lead to more efficient training and data generation, and paves the way to novel architectures integrating different generative approaches and generation tasks.

## 1 Introduction

Diffusion-based generative models, first introduced in Sohl-Dickstein et al. (2015), have seen great successes in recent years since the works of Song & Ermon (2019); Ho et al. (2020). In these models, data are transformed into noise following a diffusion process, and a transformation simulating the reverse process is learned which is then used to map noise into generated samples. In physics, the theory of renormalization group (RG) flows has been a basic tool in the study of a wide range of physical phenomena, including phase transitions and fundamental physics, both in theoretical as well as numerical approaches. In short, an RG flow prescribes a way of erasing the high-frequency information of a physical theory, while retaining the information relevant for the long-wavelength physics. As such, there are clear analogs between score-based generative models and RG flows, at least at a conceptual level. Indeed, it has been known for a long time that RG flows in quantum field theories can also be described as a diffusive process (Zinn-Justin, 2002; Gaite, 2001; Cotler & Rezchikov, 2023; Berman et al., 2023). In both cases, information gets erased along the flow and many different initial distributions get mapped into the same final distribution – a feature often referred to as "universality" in the physics literature. In the diffusion context, the "universal" distribution is given by the chosen noise distribution, independent of the data distribution.

However, there are salient differences between the ways diffusion models and RG erase information. First, the **basis**: the diffusive RG process is diagonal in the frequency-basis while the standard diffusion models typically diffuse diagonally in the pixel-basis. Second, the **prior distribution**: the endpoint of RG is a scale-invariant distribution, often with the same second-order statistics as the distribution one starts with at the beginning of the RG flow. The standard diffusion models on the other hand indiscriminately map all data distributions to that of white noise. Third, the **component-wise noising schedule**: RG flows erase information sequentially from high to low frequencies, while the original diffusion model has the same noise schedule for all pixels. In our chosen basis, we allow each component to have its own noising schedule. Given a choice of diagonal basis and prior distribution, this third aspect provides our model with its novel flexibility, which in particular enables us to continuously interpolate between autoregressive generation and standard diffusion models. These considerations lead us to the framework of generative unified diffusion (GUD) models which incorporate the freedom in design choices in the above three aspects.

Autoregressive models, such as next-token prediction models, play an increasingly dominant role in modern-day machine-learning applications such as LLMs, and seem to be distinct from diffusion models at first glance. In autoregressive models, tokens are generated one at a time, conditional on previously generated ones, while diffusion models generate information in all components simultaneously. We will show that the two can in fact be unified in our framework, which in particular allows for *soft-conditioning* generative processes. Intuitively, this means that we can condition on partial information from other components as long as that information has already been generated in the diffusion process.

To probe the impact of the different choices, in particular of different soft-conditioning schedules, and to provide a setup for numerically exploring these choices, we perform numerical experiments with a noising-state-conditioned model in §5.

## 2 RELATED WORK

**Prior Distribution.** The choice of the noise distribution was first discussed in (Song & Ermon, 2020), in which the Technique 1 involves choosing a given isotropic variance for the prior normal distribution. Non-isotropic noise was explored in (Voleti et al., 2022), however without a component-wise schedule. In this work we explain and demonstrate via numerical experiments that variance-matching noise should be accompanied by a suitably chosen component-wise schedule in order to achieve a hierarchical structure of the generative model, which is to some extent inherent in the standard diffusion model with an isotropic noise distribution.

**Basis.** The idea that diffusion models can incorporate the multi-scale nature of the dataset has inspired various models with non-standard choices of data representation, including Guth et al. (2022); Ryu & Ye (2022); Ho et al. (2022). In these works, the generation is sharply autoregressive between different hierarchies of generation (between different resolutions, for instance). Our model accommodates these data representations as special examples, and moreover allows for soft-conditioning between different hierarchies. Particularly relevant work is the Blurring Diffusion Models (Hoogeboom & Salimans, 2024), following earlier work (Rissanen et al., 2023), where the authors proposed diffusion models in the frequency basis. In (Hoogeboom & Salimans, 2024), §4.2, the authors briefly pointed out that these are special cases of a diagonalizable linear SDE without further exploring the general cases.

**Component-Wise Noising Schedule.** In (Lee et al., 2023) it was pointed out that autoregressive generation in the diffusion model framework by noising/denoising a group of degrees of freedom at a time, though the authors did not discuss how to choose such groups or the possibility to soft-condition different degrees of freedom. Explicitly sequential diffusion has been explored in (Ruhe et al., 2024) for sequences of image frames. The possibility of interpolating between the standard diffusion models and token-wise autoregressive models has also been recently explored in Chen et al. (2024) in the context of causal sequence generation, in which the authors work with a chosen token-wise noising schedule and capture the information of the (partially noised) previous tokens in latent variables, on which the denoising process depends. As opposed to these work which focus on causal sequences, our work considers arbitrary hierarchies of generation, with the flexibility to tune how sharply autoregressive the generative process is. In particular, as we demonstrate in experiments, it is possible to integrate multi-scale and spatially sequential generation processes in our framework. More generally, finding an optimal component-wise noising schedule, corresponding to a path between the prior and target distribution, remains an open question. In (Das et al., 2023), a path inspired by the shortest distance path between Gaussian distributions was proposed for image generation tasks. We discuss desirable quantative features a good component-wise noising schedule should possess, based on ideas from the physics of RG flows decribed in §3.2, and provide concrete examples of schedule improvements in our numerical experiments in §5.

**Latent Space Diffusion.** In practice, for applications with high-dimensional data, the diffusion generation often takes place in a lower-dimensional latent space (Rombach et al., 2021; Sinha et al., 2021; Vahdat et al., 2021). The freedom to choose the basis proposed in our work is not to be understood as replacing latent space diffusion. Rather, our framework can straightforwardly be used in the latent space, leading to a latent GUD model.

## 3 PRELIMINARIES

### 3.1 STOCHASTIC DIFFERENTIAL EQUATIONS

In continuous time, the general diffusion setup can be described by the following Itô stochastic differential equation (SDE):

$$d\boldsymbol{\phi} = \mathbf{f}(\boldsymbol{\phi}, t)\,dt + \mathbf{G}(\boldsymbol{\phi}, t)\,d\mathbf{w}\ , \tag{1}$$

where $d\mathbf{w}$ represents a white noise Wiener process. We use $\boldsymbol{\phi} \in \mathbb{R}^d$ to denote a vector. In the above, we have $\mathbf{f}(\,\cdot\,, t) : \mathbb{R}^d \to \mathbb{R}^d$ and $\mathbf{G}(\,\cdot\,, t) : \mathbb{R}^d \to \mathbb{R}^{d \times d}$. The reverse-time SDE is given by (Anderson, 1982)

$$d\boldsymbol{\phi} = \left(\mathbf{f}(\boldsymbol{\phi}, t) - \nabla \cdot (\mathbf{G}\mathbf{G}^T)(\boldsymbol{\phi}, t) - \mathbf{G}\mathbf{G}^T \nabla_{\boldsymbol{\phi}} \log p_t(\boldsymbol{\phi})\right) dt + \mathbf{G}(\boldsymbol{\phi}, t)d\bar{\mathbf{w}} \tag{2}$$

where $\bar{\mathbf{w}}$ is the inverse Wiener process.

The probability density $p(\boldsymbol{\phi}, t)$ corresponding to the SDE equation 1 solves the following Fokker-Planck equation (or Kolmogorov's forward equation) (Oksendal, 1992)

$$\frac{\partial}{\partial t} p(\boldsymbol{\phi}(t)) = -\sum_{i=1}^{d} \frac{\partial}{\partial \phi_i} \left(f_i(\boldsymbol{\phi}, t)p(\boldsymbol{\phi}(t))\right) + \frac{1}{2} \sum_{i=1}^{d} \sum_{j=1}^{d} \frac{\partial^2}{\partial \phi_i \partial \phi_j} \left(\sum_{k=1}^{d} G_{ik} G_{jk}\, p(\boldsymbol{\phi}(t))\right), \tag{3}$$

where $\phi_i$ denotes the component of $\boldsymbol{\phi}$ in a given basis.

### 3.2 RENORMALIZATION GROUP (RG) FLOWS

As mentioned in the introduction, the renormalization group refers to a collection of methods in physics that aim to progressively remove the high-frequency degrees of freedom while retaining the relevant low-frequency ones. In other words, one aims to remove the irrelevant details of the physical system without altering the physics at the larger scale one is interested in. By doing so, one hopes to be able to robustly calculate the universal macroscopic features of the physical systems.

There are many ways physicists have proposed to achieve this goal, starting with the seminal work of Kadanoff (1966) and Wilson (1971a;b). How to improve the understanding and the implementation of RG flows, including efforts involving machine learning methods, remains an active topic of investigation in physics. Here we consider the exact RG (ERG) formalism, a non-perturbative method pioneered by Polchinski (1984) for quantum field theories. In this RG method, one implements Wilson's idea of RG by specifying a *cutoff kernel* $K_k(\Lambda) := K(k^2/\Lambda^2)$ for a given *cutoff scale* for each frequency $k$, with the property that $K_k(\Lambda) \to 1$ when $k \ll \Lambda$ and $K_k(\Lambda) \to 0$ when $k \gg \Lambda$. With this, one erases information on frequencies much larger than $\Lambda$. One example of such cutoff kernels is the sigmoid function.

Given a physical theory and a choice of cutoff kernel, one can define physical probability distributions $p_\Lambda[\boldsymbol{\phi}]$ that satisfy a differential equation which is an infinite-dimensional version of the Fokker-Planck equation 3, where the role of diffusion time is played by $t = -\log(\Lambda/\Lambda_0)$ for some reference scale $\Lambda_0$.

### 3.3 STANDARD DIFFUSION MODELS

In diffusion-based generative models, a forward diffusion process that gradually transforms data samples into noise following a particularly simple SDE is inverted to transform noise into images. A commonly used forward SDE is the finite-variance[1] SDE (Song et al., 2021) defined as:

$$d\boldsymbol{\phi} = -\tfrac{1}{2}\beta(t)\boldsymbol{\phi}\,dt + \sqrt{\beta(t)}\,d\mathbf{w}, \tag{4}$$

where the initial vector $\boldsymbol{\phi}(0) \in \mathbb{R}^d$ represents the data sample and $d\mathbf{w}$ denotes the standard Wiener process. The function $\beta : [0, T] \to \mathbb{R}_+$ which determines the SDE is the predefined noise schedule.

---

[1]This is referred to as the "variance-preserving" (VP) diffusion in some literature. We will reserve the term to cases when the variance is actually strictly constant throughout the diffusion process, which we will discuss in §4.4.

The reverse-time SDE follows from specializing equation 2 and reads

$$\mathrm{d}\boldsymbol{\phi} = \left[ -\tfrac{1}{2}\beta(t)\boldsymbol{\phi} - \beta(t)\nabla_{\boldsymbol{\phi}} \log p_t(\boldsymbol{\phi}) \right] \mathrm{d}t + \sqrt{\beta(t)}\mathrm{d}\bar{\mathbf{w}}\,, \tag{5}$$

where $\mathrm{d}\bar{\mathbf{w}}$ is a reverse-time Wiener process, and $\nabla_{\boldsymbol{\phi}} \log p_t(\boldsymbol{\phi})$ is the score function of the marginal distribution at time $t$. The task for machine learning is thus to approximate the score function, which can be achieved by denoising score matching (Vincent, 2011) with the objective function

$$\mathcal{L}_{\mathrm{DSM}} = \mathbb{E}_{t,\boldsymbol{\phi}(0),\epsilon} \left[ \lambda(t) \big\| s_\theta(\boldsymbol{\phi}(t),t) - \nabla_{\boldsymbol{\phi}(t)} \log p_t(\boldsymbol{\phi}(t)|\boldsymbol{\phi}(0)) \big\|^2 \right]\,, \tag{6}$$

where $\epsilon \sim \mathcal{N}(0,\mathbf{I})$ is Gaussian white noise, $\boldsymbol{\phi}(t) = \alpha(t)\boldsymbol{\phi}(0) + \sigma(t)\epsilon$ is the noised data at time $t$, $\lambda : [0,T] \to \mathbb{R}_+$ is a weighting function, and $\alpha(t), \sigma(t)$, and $\beta(t)$ are functions capturing the equivalent information about the noising schedule. They are defined in equation 14 by specializing $\beta_i = \beta$ etc. Importantly, the choice of SDE (4) leads to an Ornstein-Uhlenbeck (OU) process and the conditional score $\nabla_{\boldsymbol{\phi}(t)} \log p_t(\boldsymbol{\phi}(t)|\boldsymbol{\phi}(0))$ can be computed analytically (Song et al., 2021).

## 4 METHODS

### 4.1 DIAGONALIZABLE ORNSTEIN UHLENBECK PROCESS

Returning to the general diffusion SDE (1), we now consider the special case in which $\mathbf{f} = \boldsymbol{F}\boldsymbol{\phi}$ and $\boldsymbol{F}$ is $\boldsymbol{\phi}$-independent . This guarantees that the SDE describes a Ornstein-Uhlenbeck process admitting analytical solutions for the conditional distribution $p_t(\boldsymbol{\phi}(t)|\boldsymbol{\phi}(0))$ required for denoising score matching. Moreover, we consider a choice of *simultaneously diagonalizable* $\mathbf{F}$ and $\mathbf{G}$:

$$\boldsymbol{F} = M^{-1}\tilde{\boldsymbol{F}}M, \;\; \boldsymbol{G} = M^{-1}\sqrt{\boldsymbol{\beta}} \tag{7}$$

with some constant matrix $M$ and diagonal $\boldsymbol{\beta} = \mathrm{diag}(\beta_i)$ and $\tilde{\boldsymbol{F}} = \mathrm{diag}(\tilde{F}_i)$.

In terms of the parameterization $\boldsymbol{\chi} := M\boldsymbol{\phi}$, the SDE equation 2 is equivalent to $d$ decoupled SDEs of the form

$$\mathrm{d}\chi_i = \tilde{F}_i(t)\chi_i\mathrm{d}t + \sqrt{\beta_i(t)}\,\mathrm{d}w \tag{8}$$

with the reverse SDE given by

$$\mathrm{d}\chi_i = \left( \tilde{F}_i(t)\chi_i - \beta_i(t)\nabla_{\chi_i} \log p_t(\chi) \right) \mathrm{d}t + \sqrt{\beta_i(t)}\mathrm{d}\bar{w}\,. \tag{9}$$

The choice of the transformation matrix $M$ is a choice of data representation in which the diagonal score-based diffusion based on the SDE equation 8 and equation 9 can be efficiently performed.

In particular, as the Wiener process is invariant under orthogonal transformations, it is convenient to view the change of basis (given by $M$) as the composition of an orthogonal ($U$) and a scaling ($S$) transformation: $M = S^{-1}U$. In terms of the original $\boldsymbol{\phi}$ variables, the forward SDE then reads

$$\mathrm{d}\boldsymbol{\phi} = F\boldsymbol{\phi}\,\mathrm{d}t + U^{-1}S\sqrt{\boldsymbol{\beta}}\,\mathrm{d}w = U^{-1}\tilde{\boldsymbol{F}}U\boldsymbol{\phi}\,\mathrm{d}t + \sqrt{\boldsymbol{\beta'}}\,\mathrm{d}w' \tag{10}$$

where $\sqrt{\boldsymbol{\beta'}} = U^{-1}\sqrt{\boldsymbol{\beta}}U$ and $\mathrm{d}w' = \sqrt{\Sigma_{\mathrm{prior}}}\,\mathrm{d}w$ is a Wiener process with covariance matrix $\Sigma_{\mathrm{prior}} = U^{-1}S^2 U$.

The choice of $M$, particularly the orthogonal part $U$, captures the freedom in our unified framework to choose the **basis** in which the diffusion process is diagonal. Moreover, the scaling $S$ then determines the choice of the noise (**prior**) distribution $p_{\mathrm{prior}} = \mathcal{N}(0, \Sigma_{\mathrm{prior}})$, which the forward process approaches at late times. Finally, note that $\beta_i(t)$ are a priori independent functions of $t$ for each component $i$. The choice of $\beta_i(t)$ thus captures the choice of a **component-wise noising schedule**.

Due to the diagonal property of the SDE, the denoising score matching loss function equation 6 for learning the Stein score $\nabla_{\chi_i} \log p_t(\chi)$ can be straightforwardly generalized to the GUD models:

$$\mathcal{L}_{\mathrm{GUD}} = \mathbb{E}_{t,\boldsymbol{\chi}(0),\epsilon} \sum_{i=1,\dots,d} \lambda_i(t)\Big| s_{i,\theta}(\boldsymbol{\chi}(t),t) - \nabla_{\chi_i(t)} \log p_t(\boldsymbol{\chi}(t)|\boldsymbol{\chi}(0))\Big|^2 \tag{11}$$

where the $\boldsymbol{\lambda} = (\lambda_1,\dots,\lambda_d) : [0,T] \to \mathbb{R}_+^d$ is the weighting vector. In our experiments, we let $\lambda_i(t) = \sigma_i^2(t)$, with the aim to scale the loss to be an order-one quantity and generalizing the

common weighting factor $\lambda(t) = \sigma^2(t)$ in the standard diffusion loss (6) (Song & Ermon, 2019). The SDE (9) with the learned score, when discretized, leads to a hierarchical generative model with model density

$$\tilde{p}(\boldsymbol{\chi}(0)) = \int \left( \prod_{k=1}^{T} d^d \boldsymbol{\chi}(\tfrac{k}{T}) \right) \left( \prod_{\ell=0}^{T-1} p(\boldsymbol{\chi}(\tfrac{\ell}{T})|\boldsymbol{\chi}(\tfrac{\ell+1}{T})) \right) \tilde{p}(\boldsymbol{\chi}(1)) \tag{12}$$

where $T$ is the number of steps in the discretization, and the prior distribution is given by the noise distribution $\tilde{p}(\boldsymbol{\chi}(1)) = \mathcal{N}(0, \mathbf{I})$.

## 4.2 Finite-Variance Diffusion and the Signal to Noise Ratio

In the coordinate given by $\boldsymbol{\chi}$, we now further specialize equation 8 to the following finite-variance diffusion process: with $\tilde{\boldsymbol{F}}(t) = -\frac{1}{2}\boldsymbol{\beta}(t)$, the corresponding SDE reads

$$d\chi_i = -\frac{1}{2}\beta_i(t)\chi_i dt + \sqrt{\beta_i(t)} d\mathrm{w}. \tag{13}$$

Integrating the above gives $\chi_i(t) = \alpha_i(t)\chi_i(0) + \sigma_i(t)\epsilon$, where $\epsilon \sim \mathcal{N}(0,1)$ and

$$\alpha_i(t) = \exp\left(-\frac{1}{2}\int_0^t \beta_i(s)\,\mathrm{d}s\right), \quad \text{and} \quad \sigma_i(t)^2 = 1 - \alpha_i(t)^2. \tag{14}$$

It follows that the variance

$$\mathrm{Var}(\chi_i(t)) = \alpha_i(t)^2 (\Sigma^{(\boldsymbol{\chi})}(0))_{ii} + \sigma_i(t)^2 \tag{15}$$

interpolates between 1 and the data variance $(\Sigma^{(\boldsymbol{\chi})}(0))_{ii}$, and is in particular finite at all stages of diffusion. In the above, we have used the following notation for the data covariance matrix

$$(\Sigma^{(\boldsymbol{\chi})}(0))_{ij} := \mathbb{E}_{p_{\mathrm{data}}(\boldsymbol{\chi}(0))} \left[ (\chi_i(0) - \overline{\chi_i(0)})(\chi_j(0) - \overline{\chi_j(0)}) \right], \quad \text{where} \quad \overline{\chi_i(0)} := \mathbb{E}_{p_{\mathrm{data}}(\boldsymbol{\chi}_0)}[\chi_{0,i}],$$

An important quantity signifying the stage of the diffusion process (for each component) is the time evolution of the ratio between the signal and the noise, captured by the *signal-to-noise ratio*,

$$\mathrm{SNR}_i(t) := \mathbb{E}\left( \frac{(\alpha_i(t)\chi_i(0))^2}{\sigma_i(t)^2} \right) = (\Sigma^{(\boldsymbol{\chi})}(0))_{ii} \frac{\alpha_i^2(t)}{\sigma_i^2(t)}, \tag{16}$$

where the expectation is with respect to the data and the noise distribution, and we have assumed that the data mean vanishes (which can always be made to be the case by subtracting the mean). Note that this is different from the signal-to-noise ratio quoted in some diffusion model contexts,

$$\mathrm{snr}_i(t) := \alpha_i^2(t)/\sigma_i^2(t) = e^{-\gamma_i(t)}, \tag{17}$$

as this version does not take into account the magnitude of the signal in the data. As they depend only on the schedule, we note that the functions $\beta_i$, $\alpha_i$, $\sigma_i$ and $\gamma_i$ all contain the same information.

At a given time $t \in [0,T]$ in the diffusion process, the information of $\boldsymbol{\gamma}(t) = (\gamma_1, \ldots, \gamma_d)(t) \in \mathbb{R}^d$ is what we call the *noising state*, indicating the extent to which information in the data has been replaced by noise at that time. As a function of $t$, the evolution of the noising state traces out a path connecting the data distribution $p_{\mathrm{data}}$, corresponding to

$$\boldsymbol{\alpha}(\boldsymbol{\gamma}) = (\alpha_1, \ldots, \alpha_d) = (1, \ldots, 1),$$

and the prior distribution $p_{\mathrm{prior}}$, corresponding to $\boldsymbol{\alpha} = (0, \ldots, 0)$, where $\alpha_i = \mathrm{sigmoid}(-\gamma_i)^{1/2}$ as in equation 17. The different paths correspond to different Ornstein-Uhlenbeck processes, as defined in equation 13, with different diffusion dynamics.

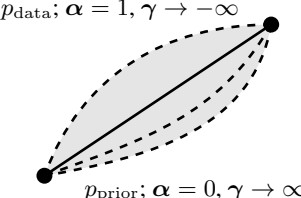

Figure 1: Different noising schedules $\boldsymbol{\gamma}(t)$.

This highlights the fact that the freedom in component-wise noising schedules in the GUD model is fundamentally larger than the freedom in the noising schedule in standard diffusion models, which is given by different choices of the function $\gamma(t)$ with $\gamma_i(t) = \gamma_j(t) = \gamma(t)$. In this case, all choices of $\gamma(t)$ trace out the same diagonal path (as long as the boundary values $\gamma(0)$, $\gamma(T)$ are held fixed) and merely amount to different time parameterizations (Kingma et al., 2023). In contrast, different component-wise schedules generically correspond to genuinely different paths, as illustrated in the schematic Figure 1 where the continuous line corresponds to the standard diffusion schedule (with any time parametrization) and the dashed lines represent other possible component-wise schedules.

### 4.3 UNIFICATION VIA SOFT-CONDITIONING

The above form of SNR clarifies an implicit hierarchical structure of the standard diffusion models: even when $\gamma_i(t) = \gamma(t)$ is identical for all components $i$, the components with larger amplitudes have larger signal-to-noise ratio $\text{SNR}_i(t) = (\Sigma^{(\boldsymbol{x})}(0))_{ii}e^{-\gamma(t)}$, and are in this sense less "noised" throughout the diffusion process. As a result, the generation process equation 12 and in particular the modeling of the probability $p(\boldsymbol{\chi}(\frac{\ell}{T})|\boldsymbol{\chi}(\frac{\ell+1}{T}))$ conditional on the previous state is implicitly a process of generating the less important features (with smaller amplitude) conditional on the more important features (with larger amplitude) that have already been partially generated. It is clear that by making more general choices of component-dependent noising schedules $\gamma_i(t)$ one can tune the degree of this *s*oft-conditioning property, as we will explore in the experiments below. In the extreme case when the support of $\beta_i(t)$ and $\beta_{i \neq j}(t)$, namely the "active time" for the $i$th resp. $j$th component, do not overlap, we arrive at autoregressive generation, in which one feature/token (or one group of features/tokens) is generated at each time, conditional on those that have been generated already. See Figure 6 for the visualization of a specific example. In this way, the freedom to choose a component-dependent noising schedule in our GUD model enables us to interpolate between standard diffusion and autoregressive generation.

### 4.4 WHITENING

A particularly interesting choice for the matrix $M = S^{-1}U$ is the orthogonal transformation $U$ that diagonalizes the data covariance matrix $\Sigma^{(\boldsymbol{\phi})}(0)$, and the diagonal matrix $S^{-1}$ that performs a *whitening* transformation. In other words, we choose $S$ and $U$ such that the data covariance matrix matches $\Sigma^{(\boldsymbol{\phi})}(0) = U^{-1}S^2U$. Note that $M$ is then precisely the familiar PCA transformation followed by a whitening transformation which makes the variance uniform. In the context of diffusive generation, such a basis has the following appealing features. First, the softness of the soft-conditioning, manifested via the evolution of the signal-to-noise ratio equation 16, is now completely controlled by the component-wise schedule $\gamma_i(t)$, which can make the design process of the diffusion modeled more streamlined and uniform across different applications with different datasets. Second, with such a choice the covariance matrix actually remains constant throughout the diffusion process as the data covariance $\Sigma^{(\boldsymbol{x})}(0) = \mathbf{I}$ is now the same as the noise covariance, and the finite-diffusion equation 13 is *variance preserving* in the strict sense. In other words, the conditional number of the covariance matrix is always one and the generative process does not need to alter the second-order statistics. We expect this property to be beneficial in some situations for learning and discretization.

### 4.5 NOISING-STATE CONDITIONAL NETWORK ARCHITECTURE

For the score network architecture, we follow the approach of predicting the noise $\epsilon$ given a noised image (Ho et al., 2020), trained via denoising score matching (Vincent, 2011). In standard diffusion models, this score network is typically conditioned on the time variable or an equivalent object such as $\gamma(t)$ (Kingma et al., 2023). The introduction of a component-wise schedule in our framework suggests generalizing this by conditioning the model on the more informative component-wise noising state, represented by the component-wise noise state $\boldsymbol{\gamma}(t) = (\gamma_1, \ldots, \gamma_d)(t)$. Since this is a vector of the same dimension as the data and not a scalar, a modification of the network architecture is required. We have implemented this by incorporating cross-attention between the data and the noising state, further details can be found in section A of the appendix.

Since any choice of the noising schedule $\gamma_i(t) = \gamma(t)$ in standard diffusion models can be thought of as just a reparametrization of time (Kingma et al., 2023) (cf. §4.2), the diffusion time $t$ itself suffices as a feature for the network to indicate the noising state. For our GUD models, this is true only for a fixed schedule choice. By conditioning directly on $\boldsymbol{\gamma}$ instead of $t$, our score network is directly conditioned on the instantaneous noising state, and not on the totality of its path, namely the schedule $\boldsymbol{\gamma}(t)$. This enables us to train a single network for a range of schedules, as we will do in the experiments described in the next section. The set of values $\boldsymbol{\gamma} \in \mathbb{R}^d$ used during training bound a region, visualized schematically by the shaded area in Fig. 1. This is implicitly the region of the values of $\boldsymbol{\gamma} \in \mathbb{R}^d$ where the score function has been learned. This suggests the possibility of using any particular path within the shaded region for generation, which might differ from the path used

for training (indicated by the dashed lines in Fig. 1). This feature of the GUD model may facilitate the numerical optimization of component-wise schedules in future work.

In our experiments, we also use a minimally modified version of the NSCN++ architecture from Song et al. (2021) with no cross-attention. We found that it is sufficiently expressive as long as we consider a low-dimensional family of schedules. We attribute that to the fact that in this case not all information encoded in $\gamma$ is needed to determine the noising state.

## 5 EXPERIMENTS

We will now showcase the flexibility of the GUD model with some examples, and conduct preliminary investigations into the effects of these different design choices on the behavior of diffusion models and their resulting sampling quality. An overview of the experiments, highlighting the relevant design choices, is given in the following table.

|  | §5.1 | §5.2 | §5.3 |
|---|---|---|---|
| basis | pixel, PCA , FFT | column | wavelet $\otimes$ column |
| prior | isotropic Gaussian and variance-matching Gaussian | isotropic Gaussian | isotropic Gaussian |
| noising schedule | varying softness and ordering variables | varying softness | varying softness |
| other applications |  | image extension |  |

### 5.1 SOFT-CONDITIONING SCHEDULES

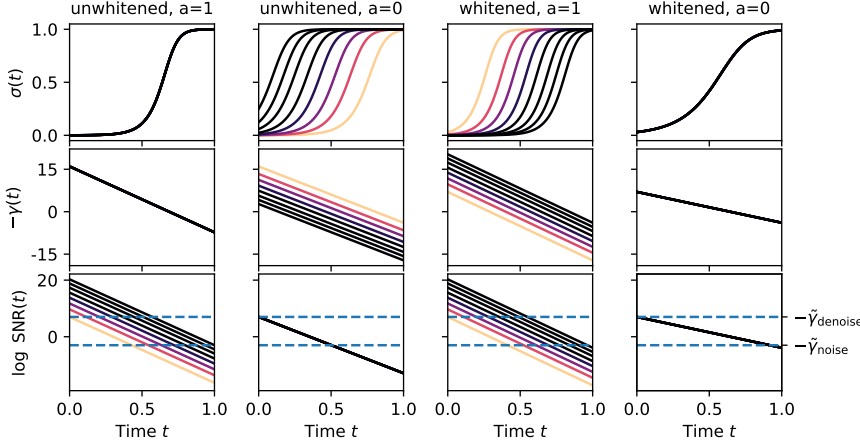

Figure 2: For eight of the PCA components $\chi_i$ of CIFAR-10, we visualize the OU noise level $\sigma_i(t)$, the corresponding noising path $\gamma_i(t) = \text{logit}(\sigma_i^2(t))$ for the linear schedule of equation 18, and the corresponding signal-to-noise ratio, for the following four choices. Blue dashed lines indicate chosen minimal noising/reconstruction levels. From left to right: (a) Standard diffusion. (b) Hierarchy-less generation with $\gamma$ chosen such that $\log \text{SNR}_i(t) = \log \text{SNR}_j(t)$. (c) With whitened data, and with $\gamma$ chosen such that $\log \text{SNR}_i(t)$ is identical to the column (a). (d) Hierarchy-less generation with whitened data.

First, we investigate the effect of choosing different bases, priors, and schedules by conducting experiments on the unconditional generation of CIFAR-10 images. We choose the simplest example of a linear noise schedules $\gamma_i(t)$ (cf. equation 18). We can then change the time weighting with a non-linear time reparametrization $\gamma_i(s(t))$, where $s$ monotonously increasing function preserving the interval $[0, T]$. In the first setup, we choose the basis, given by the orthogonal transformation $U$ described in §4.1, to be the PCA basis. For the prior, we choose our noise distributions to be either isotropic Gaussian or Gaussian with covariance matching that of the data. As explained in §4.1, the latter is equivalent to whitening the data (in the PCA basis) and using isotropic Gaussian noise. We will therefore refer to the two choices of priors as *whitened* and *unwhitened*.

In the second setup, we use the Fourier basis and consider a two-parameter family of component-wise noising schedules, where we vary the precise ordering of the different Fourier components (given by the "ordering variables") as well as the softness parameter of the soft-conditioning schedule.

**Linear component-wise schedules.**  We define our schedules with the linear functions with the same slope for different components $i$

$$\gamma_i(t) = \gamma_{\min,i} + t\Delta\gamma, \quad \text{with } t \in [0, T] = [0, 1].$$ (18)

To determine the values for $\gamma_{\min,i}$ and $\gamma_{\max,i} = \gamma_{\min,i} + \Delta\gamma$, we have the following two considerations. First, the endpoints must guarantee sufficient noising and denoising for each $i$. Secondly, the relative offset of the linear functions between components determines the ordering and level of autoregressiveness or softness of the generative process. Examples of linear schedules are visualized in Figure 2.

Denoting the variance of the component $\chi_i$ by $\Sigma_i := (\Sigma^{(\mathbf{x})}(0))_{ii}$, the signal-to-noise ratio becomes $\log \text{SNR}_i(t) = -\gamma_i(t) + \log \Sigma_i$ as discussed in §4.2. We thus define the minimal levels of denoising and noising that we require at initial and final times, respectively:

$$\tilde{\gamma}_{\text{denoise}} = -\min_i \log \text{SNR}_i(t = 0) \quad \text{and} \quad \tilde{\gamma}_{\text{noise}} = -\max_i \log \text{SNR}_i(t = 1).$$ (19)

Fixing these values defines a constraint on the schedule. Moreover, to employ the inverse process as a generative model, the distribution at the final time must also be sufficiently close to the prior normal distribution from which we draw initial samples. We therefore also require $\sigma_i(t = 1) \geq \sigma_{\min}$, translating to $\tilde{\gamma}_{\text{noise}} \geq \text{logit}(\sigma_{\min}^2) - \min_i \log \Sigma_i$.

To parameterize the hierarchical structure of the generative process, we associate an ordering variable $l_i$ to each component that determines the hierarchy between them. Like many (natural) image datasets, CIFAR-10 is an example of what might be called frequency-based datasets, by which we mean datasets with a natural meaning of locality, whose covariance is approximately diagonalized in the Fourier basis and whose variance is generally decreasing with increasing frequencies (Tolhurst et al., 1992; Field, 1987). For these datasets, the hierarchical structure can naturally be specified in terms of the related notions of variance, frequencies, and resolution, as familiar from image processing. We choose our ordering variable $l_i$ to capture this notion of hierarchy.

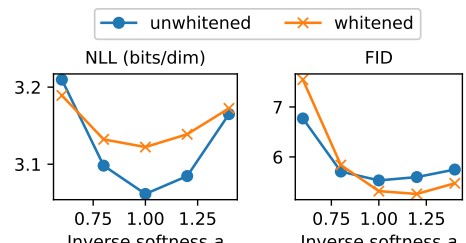

Figure 3: Dependence of model quality in terms of negative log-likelihood (left) and FID (right) on the softness parameter for the linear schedule in §5.1. The schedule is defined in PCA components and results are shown both for unwhitened and whitened data scaling (i.e. white and data-matching priors). Training on CIFAR-10 using a single score-network for each choice of scaling. Standard diffusion corresponds to $a = 1$ in the unwhitened case.

To parametrize the level of autoregressiveness of the soft-conditioning linear schedule, we introduce a parameter $a > 0$ by which we multiply the $l_i$. Together they determine the slope and the offset in the following way. Letting $l_{\max} = \max_i l_i$ and $l_{\min} = \min_i l_i$, we set

$$\begin{aligned} \gamma_{\min,i} &= \tilde{\gamma}_{\text{denoise}} + \log \Sigma_i + a(l_i - l_{\max}) \\ \Delta\gamma &= \tilde{\gamma}_{\text{noise}} - \tilde{\gamma}_{\text{denoise}} + a(l_{\max} - l_{\min}) \end{aligned}$$ (20)

The larger $a$ is, i.e. the smaller the softness $1/a$, the more autoregressive the schedule becomes. Conversely, in the limit of extreme softness (small $a$) the hierarchical nature of the generative model disappears. The parameters of our linear schedules are thus the ordering variables $l_i$, the parameter $a$, and the SNR endpoints given by $\tilde{\gamma}_{\text{denoise}}, \tilde{\gamma}_{\text{noise}}$.

**Softness in PCA space.**  In the first experiment, we apply the above linear schedule in the whitened and unwhitened PCA bases. We choose the ordering variable to be given by $l_i = -\log \Sigma_i$, which allows for the trajectory of the signal-to-noise ratio of the standard diffusion model to be reproduced at $a = 1$, also when the prior has been changed to have the same covariance as the data (see Fig. 2 (c)). We trained a single score network for the (inverse) softness parameter in the range $a \in [0.4, 1.6]$

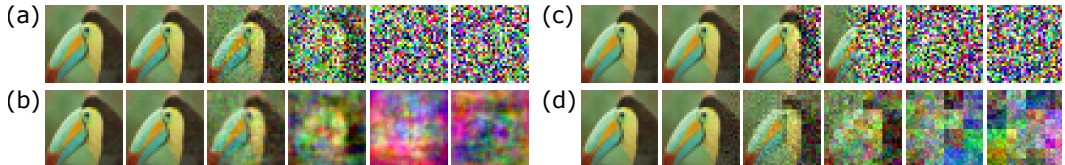

Figure 4: Diffusion forward process for a single image of CIFAR-10: (a) standard diffusion, (b) variance-matching Gaussian noise with same SNR as standard diffusion, (c) column-wise sequential schedule of §5.2 with $b = 0.5$, (d) combination of Haar wavelet and column-sequential schedule of §5.3 with $a = 0.5$, and with variance-matching Gaussian noise.

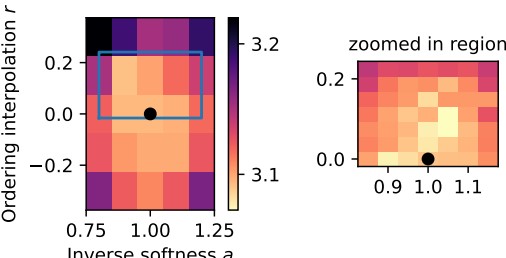
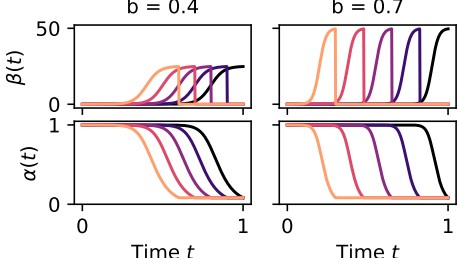

Figure 5: The model quality with a two-parameter family of schedules controlling the softness and the ordering parameters. The right figure is the region in the box on the left and the same color map is shared. The black dots indicate the parameters corresponding to standard diffusion models.

Figure 6: An illustration of column-wise schedule of §5.2 for 5 columns and different softness parameter $b^{-1}$. The larger $b$, the more autoregressive the model is, as the overlap of the "active" times with noising rate $\beta_i > 0$ decreases, and similarly for the suppression factors $\alpha_i$.

by randomly sampling $a$ at each training step. Figure 3 shows the negative log-likelihood (NLL) and FID evaluated for different values of $a$. Interestingly, the unwhitened configuration performs better when measured by NLL, but worse in terms of FID, with the standard diffusion setup ($a = 1$) appearing close to optimal. See section B of the appendix for further experimental details.

**Ordering variables.** We also perform experiments in the Fourier (FFT) basis, for which the RG physics reviewed in §3.2 naturally suggests an ordering of noising based on the frequency $|k_i|$. To test the dependence of the quality of the model on ordering parameters, we consider ordering variables $l_i = (1 - r)(-\log \Sigma_i) + r(|k_i| + \delta)/\kappa$, parametrized by $r \in [0, 1]$ and interpolating between $l_i = -\log \Sigma_i$ as in the PCA experiments and the frequencies $|k_i|$. The slope and offset parameters, $\kappa$ and $\delta$, are chosen such that the range of $l_i$ are the same at $r = 1$ as at $r = 0$. We trained a score network for a range of values of $r$ in addition to the softness parameter $a^{-1}$ on CIFAR-10, with evaluation results shown in Figure 5. We find the optimal performance in terms of NLL is located slightly away but close to standard diffusion. In this experiment, we choose the prior to be the isotropic Gaussian (i.e. unwhitened).

## 5.2 SEQUENTIAL GENERATION IN REAL SPACE

While the previous experiment explores the GUD model in the context of multi-scale hierarchical generation, it can equally be applied to perform sequential generation in pixel space, as we will now demonstrate with a soft-conditioning column-wise generation model. Grouping the components according to their column in the pixel space of size $L \times L$, we index the schedules according to the columns labeled by $i = 1, \ldots L$. In the experiments we use component-wise linear schedules equation 21 similar to the ones described in §5.1, with the parameter $b$ controlling the degree of softness/autoregressivity. Besides the soft-conditioning schedules in §5.1, this example also serves as a demonstration of how GUD model is capable of interpolating between standard diffusion and autoregressive generation.

**Training on PCAM dataset.** We trained separate score networks at $b = 0.3$ and $b = 0.5$ on the PCAM dataset (Veeling et al., 2018), downscaled to $32 \times 32$ pixels, obtaining negative log-likelihoods of $3.90$ and $3.94 \, \text{bits/dim}$, respectively.

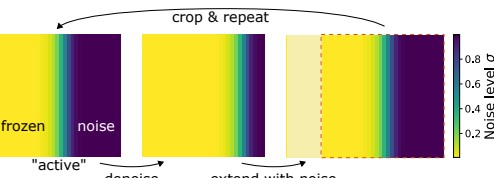

Figure 7: Left: Images generated with the column-wise schedule and a score network trained on square data. Right: Visualization of the soft column-wise generation applied to image extension. The yellow part of the image has already been generated and is hence "frozen" and the blue region corresponds to the part of the image that is yet to be generated. The middle part with green color is the active region where $\beta_i \neq 0$.

**Image extension.** The column-wise schedule can be employed to extend images in the following iterative way. We use the schedule with the property that the SDE is frozen, i.e. $\beta_i \approx 0$, except for a subset of columns, corresponding to a range of $i$ and colored in green hues in the right panel of Figure 5.2, that are being actively noised or denoised. After the active region sweeps from left to right through the whole square, the process can be repeated by sliding the constant-sized score network to the right to generate an extension of the image. The locality property of the image makes it possible to generate a new column depending only on a subset of its left-side neighbors which fit into the truncated input of the constant-sized score network. The left panel of Figure 5.2 shows three examples of image strips generated in this manner, using the score network trained on $32 \times 32$ images at $b = 0.5$. More details can be found in section C of the appendix.

### 5.3 HAAR WAVELETS

To further showcase the versatility of our unified framework, we integrate Haar wavelet decomposition with a column-wise noise schedule among the wavelet components at each hierarchical level. This extends the wavelet-conditioned score matching of Guth et al. (2022) by including a parameter allowing for soft-conditioning, and incorporating column-wise sequential noising at each level.

Concretely, we use two parameters $a$ and $b$ to parametrize the (inverse) softness among the different levels of wavelet components and the columns within each level, respectively, parametrizing a linear schedule equation 22 similar to the one in §5.1.

We trained a score network for $a \in [0.3, 0.7]$ and $b \in [0.3, 0.7]$ on CIFAR-10 for 300k steps and using $\mathcal{N} = 3$ hierarchical levels, with the linear schedule given in equation 22. See Figure 4(d) for a visualization of this schedule. Similar to the results in §5.1, the model quality again depends on the softness parameters, with the lowest NLL value reached being $3.17 \, \text{bits/dim}$.

## 6 CONCLUSIONS

In this work, we proposed the GUD framework, which naturally integrates novel design freedoms in diffusion-based generation. Notably, the framework eliminates the rigid boundary between diffusive and autoregressive generation methods and instead offers a continuous interpolation between the two. This flexibility paves the way for a broad range of potential applications.

First, our experiments indicate that choices in all three aspects we investigate in the present work – the diagonal basis, the prior distribution, and the component-wise schedule – do have an influence on the final quality of the model. As a result, there is potentially vast room to improve the quality of diffusion models. In future work, we will address the question of the optimization of these design choices.

Second, the flexibility of our framework enables seamless integration of various approaches to generative models. For instance, we illustrated in §5.3 the possibility to combine hierarchical generation (in the wavelet basis) with sequential generation, and in §5.2 how our framework can readily be used to extend images. Similarly, the inpainting, coloring, upscaling, and conditional generation tasks can all be realized and generalized within the GUD framework, via an appropriate choice of basis and component-wise schedules.

While the scope of our numerical experimentations and our ability to optimize important hyperparameters has been limited by the compute resources available to us, we believe our theoretical framework has the potential to lead to more efficient diffusion models, a wide range of applications, and novel architecture designs.

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

## A  Noising state conditional score network architecture

Inspired by techniques in conditional generation tasks (Rombach et al., 2021), we introduced a cross-attention mechanism between the intermediate embeddings of the image and the component-wise noising state $\gamma$, allowing the network to effectively modulate its predictions based on the noising state at each stage of the diffusion process. Otherwise, we follow a U-Net architecture similar to Song et al. (2021),

To incorporate additional information on the structure of the data, we first concatenate the noising state $\gamma$ with position labels specific to the application and the choice of basis. For instance, for the PCA example in §5.1 this is the negative logarithm of the variance of each component, which is also used as the ordering variable. For the experiment in Fourier space, we used the FFT frequency label $|k|$. In §5.2-5.3 we used a sequence of integers which increments by one for each subsequent column and adjacent group of Haar wavelet components, respectively. The concatenated inputs are then processed through MLP-Mixer layers to facilitate learned embeddings of $\gamma$. Along the depth of the U-Net, a single dense layer is used to reduce the spatial extent to that of the coarse images at that level, before they are input into the cross-attention.

## B  Experimental Details

**PCA and Fourier bases.**  In our experiments we make use of datasets of colored images, which have pixel and color channel indices. Among our choices of basis is the Fourier basis. The Fourier transform (specifically the fast Fourier transform) is applied independently in each color channel. To have an analogous PCA basis, we have decided to perform the same orthogonal transformation – the one corresponding the PCA basis of the color-averaged data – in each color channel. It could be interesting to investigate further choices, including the PCA transformation that mixes the color channels.

**Linear schedules for sequential generation and wavelet basis.**  In what follows we first record the component-wise noising schedule used in sequential generation experiments described in §5.2. With the parameter $b$ controlling the varying degree of softness/autoregressivity, we define the following scheule

$$\gamma_i(t) = \text{clip}_{\gamma_{\min},\gamma_{\max}}\left(\gamma_{\min} + (t - t_i)\frac{\gamma_{\max} - \gamma_{\min}}{1 - b}\right), \quad \text{with } t_i = b\frac{L - i}{L - 1}. \tag{21}$$

The clipping, defined by the clipping function $\text{clip}_{y,z}(x) := \max(y, \min(z, x))$, has the effect of freezing the columns when the designated noising ($\gamma_{\max}$) or reconstruction level ($\gamma_{\min}$) is reached.

Next we record the component-wise noising schedule used in experiments with the wavelet basis, described in §5.3. Suppose there are $\mathcal{N}$ hierarchical levels of wavelet decompositions, labeled by $i = 1, \ldots, \mathcal{N}$, and there are $L_i$ columns in the $i$-th level, indexed by $j = 1, \ldots, L_i$, we define the offsets

$$c_i = a\frac{\mathcal{N} - i}{\mathcal{N} - 1}, \quad c_{ij} = b\frac{L_i - j}{L_i - 1}.$$

With this, we specify the linear schedule for $a, b \in [0, 1]$ to be

$$\gamma_{ij}(t) = \text{clip}_{\gamma_{\min},\gamma_{\max}}\left(\gamma_{\min} + (\gamma_{\max} - \gamma_{\min})\frac{t_i - c_{ij}}{1 - b}\right), \tag{22}$$

where $t_i = \text{clip}_{0,1}(t - \frac{c_i}{1-a})$.

**Training.**  Unless specified otherwise, training was done with a batch size of 128 using the Adam optimizer with a learning rate of $5 \times 10^{-4}$. The validation parameters used to evaluate sample quality are exponentially moving averages updated at a rate of $0.999$. The diffusion times for denoising score matching are sampled uniformly in $[0, 1]$, and schedule parameters (where applicable) were drawn uniformly from the specified range for each training batch of samples.

The score networks for both CIFAR-10 and PCAM were trained for 300k training steps on NVIDIA A100 and H100s.

Except for the scan over $a$ shown in Figure 3, in all experiments we fix $\tilde{\gamma}_{\mathrm{denoise}} = -7$ and $\tilde{\gamma}_{\mathrm{noise}} = \max[3, \mathrm{logit}(\sigma_{\min}^2) - \min_i \log \Sigma_i]$ with $\sigma_{\min} = 0.99$. For these, we use our score network architecture with cross attention between image and $\boldsymbol{\gamma}$ as described in section A.

For the NLL and FID scan of Figure 3, we obtained better training results with $\tilde{\gamma}_{\mathrm{denoise}} = -3$ and using a minimally modified version of the NSCN++ architecture of Song et al. (2021). Using the positional embedding of this architecture, the diffusion time $t$ is mapped to an embedding vector of a fixed dimension. We use the same embedding starting with $\frac{1}{N} \sum_i^N \gamma_i$ as a proxy for the time, since it is a monotonous function of $t$, and concatenate in addition with a ResNet embedding of $\boldsymbol{\gamma}$. The rest of the model architecture is unmodified. In addition, we define a non-linear time parametrization via $s(t) = \frac{t^\alpha}{t^\alpha + (1-t)^\alpha}$ such that the schedule becomes $\boldsymbol{\gamma}(s(t))$. Although we have not tested the choices of $\alpha$ exhaustively, we obtain a qualitatively similar shape as the cosine schedule of Nichol & Dhariwal for $\alpha = 0.5$. We used this value for training and sampling to obtain the results in Figure 3.

**Dataset processing.** We have used a uniformly dequantized version of the dataset, both for training and evaluation, by first adding uniform noise to each quantized pixel value and then rescaling it to $[-1, 1]$. We have additionally removed the empirical mean of the dataset, computed on all training data. Otherwise, the mean would have to be taken into account when defining the magnitude-sensitive SNR, instead of just the variances as discussed in §4.2, and dividing by the mean when "whitening" could lead to extremely large values when the variance of a component is much smaller than its mean.

**Scores representations.** The scores in different data representations, e.g. the original data $\phi$ and the chosen components $\boldsymbol{\chi}$ in the notation of §4.1, are related by

$$\nabla_{\boldsymbol{\chi}} \log p_t(\boldsymbol{\chi}) = SU^\dagger \nabla_{\boldsymbol{\phi}} \log p_t(\boldsymbol{\phi}) , \tag{23}$$

and we can always go back-and-forth between the scores in both basis.

As we base our architecture on the commonly used convolutional U-net architecture as in (Song et al., 2021), which implicitly assumes the locality and approximately shift-symmetric properties, we let the inputs and outputs of the score network to be always represented in the original image space and not the chosen PCA, Fourier, or other basis.

**Evaluation and sampling.** Evaluation of the negative log-likelihood was done using the ODE corresponding to the reverse SDE equation 9. Specifically, the SDE of equation 13, i.e.

$$d\chi_i = -\frac{1}{2} \beta_i(t)\chi_i \mathrm{d}t + \sqrt{\beta_i(t)} \, \mathrm{dw} \tag{24}$$

has a corresponding deterministic ODE that produces the same marginal probabilities, given by

$$d\chi_i = -\frac{1}{2} \beta_i(t)(\chi_i + \nabla_{\chi_i} \log p_t(\boldsymbol{\chi})) \, \mathrm{d}t . \tag{25}$$

We use the above ODE to compute the log-likelihoods of the data under the trained models, with the score above replaced by its learned approximation (for more details we refer to Song et al. (2021)). We used 6144 samples in each computation, averaging over 3 different slices of the Hessian in Hutchinson's trace estimator for each sample. For the integrator, we used Tsitouras' 5/4 method as implemented in Kidger (2021) with adaptive step size and both relative and absolute tolerance of $1 \times 10^{-4}$. To generate samples for the FID evaluations we use the Euler–Maruyama method with 1000 steps, discretizing the SDE form of the reverse process.

## C    REPEATED COLUMN-WISE GENERATION

The real-space sequential column of §5.2 generates images conditional on the left part of the image that has already been denoised, after an initial stage in which the first columns get denoised. This immediately suggests an application in reconstructing an image that is only partially available. Figure 8 shows how partially noised images can be reconstructed by filling in the right hand side of the image. Different choices of random key then generate slightly different completions.

The linear column-wise schedule in equation 21 is defined such that by integrating the diffusion process by a time $\Delta t = b/(L-1)$, the "noising front" is effectively moved by one pixel. In other words, in the forward process, a particular column after this time has reached the same SNR its left neighbor had at the previous time. Starting from an image at noising time $t > 0$, we can generate in principal infinitely long strips of texture. First, we denoise using the learned score from $t$ to $t + k\Delta t$ with $k < L$ a positive integer. Then, we cut off the first $k$ columns and store them for the final image. Next, we append $k$ columns of noise drawn from the prior to the right of the image. As long as the softness parameter $b$ was chosen sufficiently large given the particular $k$, this effectively restores the image to the noising state at the original time $t$. This process can thus be repeated, and by concatenating the previously generated left-side columns, a connected rectangular stripe of image is constructed. As an example, in Figure 5.2 we show the results with $b = 0.5$ and $k = 9$. Finally, note that this procedure only works if the training data is approximately translationally invariant.

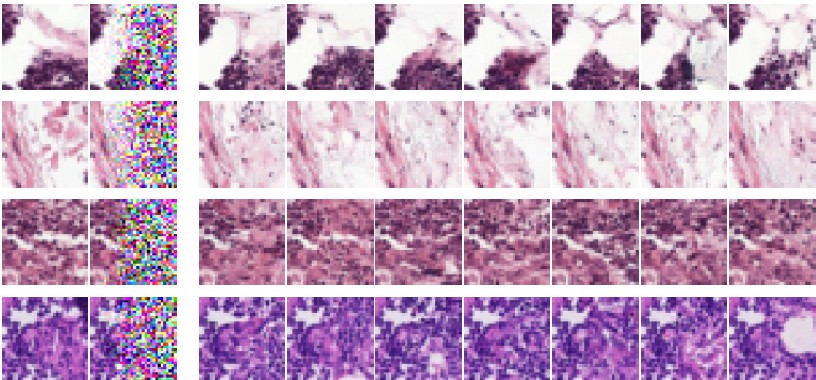

Figure 8: Reconstruction of images from the test set (left column) partially noised to $t = 0.5$ (second column) using a sequential schedule in real space as described in §5.2. The different reconstructions shown on the right differ by random key used.

## D  HAAR WAVELETS

The 2D Haar wavelet transform decomposes an image $X \in \mathbb{R}^{N \times N \times C}$ into low- and high-frequency components across multiple scales. We apply the same wavelet transform to each color channel $c = 1, \ldots, C$ (for us $C = 3$) separately. Therefore, to ease the notation we will suppress the color index in what follows. The wavelet transform at level $n$ is defined recursively as follows:

**1. Row Transformation**  Apply the 1D Haar transform along the rows:

$$L_{i,k}^{(n)} = \tfrac{1}{\sqrt{2}} \left( X_{2i,k}^{(n-1)} + X_{2i+1,k}^{(n-1)} \right),$$

$$H_{i,k}^{(n)} = \tfrac{1}{\sqrt{2}} \left( X_{2i,k}^{(n-1)} - X_{2i+1,k}^{(n-1)} \right),$$

where $i = 0, \ldots, \frac{N}{2^n} - 1$ and $k = 0, \ldots, N - 1$.

**2. Column Transforms**  Apply the 1D Haar transform along the columns to the results of the row transformation:

$$
\begin{aligned}
LL_{i,j}^{(n)} &= \tfrac{1}{\sqrt{2}} \left( L_{i,2j}^{(n)} + L_{i,2j+1}^{(n)} \right), \\
LH_{i,j}^{(n)} &= \tfrac{1}{\sqrt{2}} \left( L_{i,2j}^{(n)} - L_{i,2j+1}^{(n)} \right), \\
HL_{i,j}^{(n)} &= \tfrac{1}{\sqrt{2}} \left( H_{i,2j}^{(n)} + H_{i,2j+1}^{(n)} \right), \\
HH_{i,j}^{(n)} &= \tfrac{1}{\sqrt{2}} \left( H_{i,2j}^{(n)} - H_{i,2j+1}^{(n)} \right),
\end{aligned}
\tag{26}
$$

where $j = 0, \ldots, \frac{N}{2^n} - 1$.

**3. High-Frequency Component:** Stack the high-frequency sub-bands into a single high-frequency array at level $n$:

$$HF^{(n)} = \text{concat}\left(LH^{(n)}, HL^{(n)}, HH^{(n)}\right).\tag{27}$$

**4. Recursive Decomposition** The low-frequency component $LL^{(n)}$ becomes the input for the next level:

$$X^{(n)} = LL^{(n)}.\tag{28}$$

At each level, the transform produces one array of low-frequency components $LL^{(n)}$ and one array of high-frequency components $HF^{(n)}$. This process can be recursively applied up to a desired depth $\mathcal{N}$, resulting in a hierarchical decomposition of the image.

After level $\mathcal{N}$, the original image is represented by one lowest-frequency array and $\mathcal{N}$ higher-frequency arrays.

For example, for level 3 one obtains three high-frequency arrays $HF^{(1)}$, $HF^{(2)}$, $HF^{(3)}$, and one coarse array $LL^{(3)}$. To accommodate images with multiple color channels $C$, the transform is applied independently to each channel, and the resulting components are concatenated along the channel dimension. The factors of $\sqrt{2}$ make sure that the transform is an orthogonal transformation, whose inverse can be computed analogously.

## E   OTHER EXPERIMENTS

As suggested by one of the reviewers, we have conducted an exploratory numerical experiment for the CelebA-HQ dataset (resized to $256 \times 256$) to investigate the effect and applicability of the GUD framework to larger-sized datasets. We have trained a score-conditioned network, based on the NSCN++ architecture and enhanced with a ResNet embedding of $\gamma$ and for the linear $\gamma$ schedule described in section 5.1, parametrized by a single parameter $a$ controlling the autoregressiveness, as defined in equation 20. Here we define the components in terms of the Fourier (FFT) basis.

While a rigorous numerical investigation is beyond our resource- and time-constraints, we have made notable qualitative observations. We observed that training was improved if we use a prior that matches the power spectrum of the data. Figure 9 shows samples generated for a fixed random key and using the range of autoregressiveness parameters $a \in [0.7, 1.4]$ that we trained the score network for. The top row of the image shows samples generated using the deterministic inverse ODE flow, while the bottom row uses the inverse SDE. We observe samples generated with lower values of $a$, i.e. less autoregressive schedules, to be softer and have more residual noise. In contrast, larger $a$ appear to lead to sharper images with more artifacts. In the case of the ODE, the features of the generated faces appear more stable over choices of $a$, whereas there are visible changes in the case of the SDE.

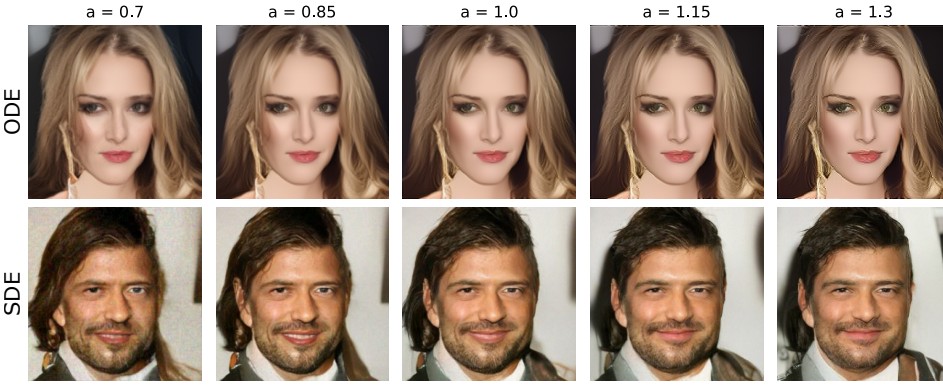

Figure 9: Samples generated at a fixed, selected random key with the inverse ODE (top row) and SDE (bottom row) flow, using a single $\gamma$-conditional score network for a range of levels of autoregressiveness $a \in [0.7, 1.3]$.

