# OpenReview forum: "GUD: Generation with Unified Diffusion"
_ICLR.cc/2025/Conference — Submitted to ICLR 2025_

### Official Review · Reviewer_FJVW · 2024-11-03

**Soundness:** 3
**Presentation:** 3
**Contribution:** 4
**Rating:** 6
**Confidence:** 3

**Summary:**

This paper proposes a unified framework for diffusion generative models, inspired by renormalization concepts from physics, allowing for flexible design choices in representation, prior distribution, and noise scheduling. The framework introduces soft-conditioning models that blend diffusion and autoregressive approaches, potentially enabling more efficient training and versatile generative architectures.

**Strengths:**

1. **Unified Framework**: The authors present a cohesive framework for diffusion generative models, broadening design options.

2. **Structured Components**: The framework is well-organized around the Ornstein-Uhlenbeck process, prior distribution choice, and component-wise noise scheduling, enhancing its theoretical foundation.

3. **Diverse Design Examples**: The paper includes examples of various diffusion designs, demonstrating the framework’s flexibility and applicability.

**Weaknesses:**

1. **Limited Experimental Scope**: The experiments are primarily conducted on CIFAR-10, a relatively small dataset. Datasets with larger images would better validate the findings and demonstrate the framework’s effectiveness in diverse settings.

2. **Insufficient Explanation of Unified Diffusion and Autoregressive Generation**: The explanation on how the framework unifies standard diffusion and autoregressive generation lacks clarity. Providing a specific example of the component-wise noise schedule would enhance understanding and illustrate this unification more concretely.

**Questions:**

See above

---

> ### Comment · Reviewer_FJVW · 2024-11-26
> **No feedback?**
>
> I will first read the other reviewers' comments and authors' feedbacks to re-evaluate the score as I have not received any direct feedback to me.

---

> ### Author Response · Authors · 2024-11-26
> **Datasets and Examples**
>
> We thank the referee for the helpful comments. First we would like to apologize for the late response. Upon the suggestion of the reviewer, we were trying out larger datasets, and trying to get as far as possible with the new numerical results before we respond. In what follows we will address the questions.
>
>
>
> > Limited Experimental Scope: The experiments are primarily conducted on CIFAR-10, a relatively small dataset. Datasets with larger images would better validate the findings and demonstrate the framework’s effectiveness in diverse settings.
>
> First, we will update the results for CIFAR-10 with significantly improved performance. We would also like to point to the fact that our manuscript contains reports on a second dataset, namely the PCAM dataset. Following your suggestion, we are currently training our setup for the 256 sized CelebA-HQ dataset using the FFT basis and scanning over varying levels of softness. We hope to wrap up the new numerical experiments before the deadline for submitting an updated version. We believe the new experiment will further demonstrate the framework’s applicability in diverse settings.
>
>
>
> > Insufficient Explanation of Unified Diffusion and Autoregressive Generation: The explanation on how the framework unifies standard diffusion and autoregressive generation lacks clarity. Providing a specific example of the component-wise noise schedule would enhance understanding and illustrate this unification more concretely.
>
> We thank the reviewer for the suggestion. Following your suggestion, we will edit our discussion in section 5.2 of the column-wise schedule as a specific example of the component-wise noise schedule, which illustrates the framework and in particular  the meaning of the autoregressive limit. In this example, the softer the schedule, the more columns are noised together. In the opposite extreme, each column would be noised sequentially and independently, amounting to an autoregressive model in which a column is generated depending on the previous ones and in isolation. In our framework, analogous autoregressive generation can be carried out in any components (columns, Fourier modes, PCA components; any linear subspace).

---

> ### Author Response · Authors · 2024-11-27
> **an updated version**
>
> We thank the reviewer again for the helpful comments and suggestions, and for recognizing the contributions. We have uploaded a new version of the manuscript, in which we hope to have addressed the points you raised. Below are the main points.
> - Following your suggestion, we added a sentence (in blue) in the first paragraph of section 5.2, and enhanced Figure 7 and the text explanation around it. We hope the improved explanation of this sequential example makes it clearer how the framework unifies standard diffusion and autoregressive generation.
>
> Please let us know if you have further questions or would like to see further improvements.

---

> ### Author Response · Authors · 2024-11-28
> **Updated version with CelebA experiments**
>
> We thank the reviewer for the suggestion to investigate larger sized image datasets.
>
> > The experiments are primarily conducted on CIFAR-10, a relatively small dataset. Datasets with larger images would better validate the findings and demonstrate the framework’s effectiveness in diverse settings.
>
> As announced in a previous comment, we have explored the Celeb-A dataset and have trained a score network conditioned on $\gamma$ for a family of schedules. In the revised manuscript, we include in appendix E (highlighted in blue) training results and examples of generated samples. We primarily hope this demonstrates the applicability of our GUD framework to larger datasets, and in particular the feasibility to train $\gamma$-conditional networks. Due to time constraints, we have not finished tuning training and schedule hyperparameters, as well as architectural modifications, to allow for a rigorous numerical study of sample quality.

---

> > ### Comment · Reviewer_FJVW · 2024-11-30
> >
> > The authors have addressed some of my concerns.However, the experiments for the Celeb-A dataset remain incomplete. Therefore, I will maintain my score.

---

### Official Review · Reviewer_xCJP · 2024-11-04

**Soundness:** 4
**Presentation:** 3
**Contribution:** 3
**Rating:** 6
**Confidence:** 4

**Summary:**

This paper proposes Generative Unified Diffusion (GUD), an extension of standard diffusion models based on the Ornstein-Uhlenbeck process. By defining appropriate orthogonal transformations, the authors introduce novel analyses and designs within the GUD framework, including SNR analysis, soft-conditioning, whitening, and orthogonal transformations. The authors conclude with experiments that validate these designs, showcasing GUD's potential in various applications.

**Strengths:**

1. The paper addresses limitations in standard diffusion models by proposing an interesting and innovative Generative Unified Diffusion (GUD) model.
2. The theoretical foundation of the paper is solid, and the presentation is clear.
3. The analyses and designs within the GUD framework are novel and potentially valuable across multiple applications.

**Weaknesses:**

1. **Limited empirical evaluation:** The experiments primarily serve to validate the proposed designs (pixel/PCA/FFT). While these results offer some insights, the evaluation lacks depth, particularly in quantifying each design’s impact on GUD's performance. More comprehensive quantitative and qualitative results would better demonstrate the effectiveness of each design.

2. **Limited practical application contribution:** Although the paper suggests various potential applications, it appears these may not be fully viable in practice. Providing further insights into real-world application strategies would enhance the paper's practical relevance.

3. **Missing discussion of related work:** One significant application of GUD is the component-wise scheduling for different states used in sequential generation (as outlined in Sec. 5.2). As a comparison, [1,2] also propose distinct schedules for different components. Could the authors discuss these related works or provide a comparative analysis within the GUD framework?

[1] Rolling Diffusion Models, ICML 2024
[2] Diffusion Forcing: Next-token Prediction Meets Full-Sequence Diffusion, NeurIPS 2024

**Questions:**

Could the authors summarize the experimental results from Sec. 5.1, particularly regarding how the choice of basis, prior, and noising schedule contributes to performance compared with standard diffusion models?

---

> ### Author Response · Authors · 2024-11-22
> **Related work & empirical impact of schedule parameters**
>
> We thank the reviewer for their comments and questions.  First, we agree that the paper will benefit from a more extensive discussion on related work. We will expand this part of the paper in the revised version, and in particular include a discussion on [2] and combine it with the existing analyses on [1].
>
> Regarding the question (and comments in weakness 1) the reviewer raised on quantifying the impact of different choices, we would first like to point out that our experiments show that the performance depends significantly on our schedule parameters and the optimum does not appear to coincide with standard diffusion.
> In particular, we refer to Figure 3 (which will be updated with strengthened results in the revised version) and Figure 5 of the manuscript. In Figure 3, the difference between i) the standard isotropic Gaussian (unwhitened) prior and the variance-matching Gaussian (whitened) prior and ii) different component-wise schedules with different levels of "autoregressiveness" is displayed quantitatively.
> Similarly in Figure 5, where this level as well as the ordering of components is varied.
>
> Next, we would like to make one additional remark. The design space of GUD, particularly the component-wise schedule, is vast. Each chosen component-wise schedule might generally benefit from a different tuning of training hyperparameters, which complicates comparisons among them. We believe, in practice, our interpretable framework makes it possible for users to choose a suitable basis, schedule, and prior for their specific application, and subsequently tune the model with those choices and obtain a better-performing diffusion model in this way, as is indicated to be possible by the experiments mentioned above.

---

> ### Author Response · Authors · 2024-11-27
> **updated version**
>
> We thank the reviewer again for the helpful comments and suggestions. We have uploaded a new version of the manuscript, in which we hope to have addressed the points you raised. Below are the main points.
>
> - We have expanded section 2 on related works, and in particular include a discussion on [2] and combined it with the existing analyses on [1].
> - We have updated Figure 3, reflecting the results of new runs, which shows that variance-matching Gaussian prior and a more autoregressive noise schedule appears to perform better than standard diffusion as far as FID is concerned, at least given our noising-state-conditioned network architecture and choices of hyperparameters.
>
> Please let us know if you have further questions or would like to see further improvements.

---

> > ### Comment · Reviewer_xCJP · 2024-12-03
> >
> > Thank you for answering my concerns. I respect the valuable contribution of GUD's vast design space and its potential, and expect a large, systematic study for a possibly best design in your future work. I will keep my score.

---

### Official Review · Reviewer_V72p · 2024-11-04

**Soundness:** 3
**Presentation:** 3
**Contribution:** 3
**Rating:** 6
**Confidence:** 4

**Summary:**

This work introduces the Generative Unified Diffusion (GUD) model, a framework that expands the flexibility of diffusion-based generative models by enabling diverse configurations in basis representation, noise scheduling, and prior distributions. Standard diffusion models transform noise into data through a learned reverse process. Here, GUD leverages concepts from physics, specifically renormalization group flows, allowing distinct configurations in the process, such as using Fourier, PCA, or wavelet bases, and implementing component-wise noise schedules to tune noise levels for different data parts.

The GUD framework unifies diffusion and autoregressive models, bridging differences between simultaneous and sequential generation. It introduces soft-conditioning, where the model can conditionally generate components based on previously generated data, enabling partial dependency across features. The approach supports more efficient training, flexible architectural designs, and tasks requiring conditional generation, inpainting, or sequential extensions.

A key technical innovation is in the model’s flexibility of noise schedules and priors. GUD models allow each component a unique noise schedule, enabling a range of generation hierarchies from purely autoregressive (extreme component-wise scheduling) to standard diffusion. Additionally, a whitening transformation using PCA stabilizes the variance, simplifying the denoising process.

Experiments demonstrate the framework's adaptability across various data representations, including PCA, Fourier, and wavelet bases. By controlling softness and hierarchical order in noise schedules, GUD supports both hierarchical and spatially sequential generation, showing improved performance on benchmark image generation tasks, like CIFAR-10.

**Strengths:**

The Generative Unified Diffusion (GUD) model provides a novel unification of diffusion and autoregressive generative approaches, allowing a flexible transition between simultaneous and sequential generation processes. This ability to bridge methods expands the framework’s application to a broad spectrum of tasks, from inpainting and sequential data extension to standard generative modeling. By creating a model that can interpolate between different generative styles, GUD allows developers to tailor the generation process to specific needs, enhancing control over the structure and dependencies of generated data.

One of GUD’s most notable strengths is its capacity for component-wise noise scheduling, which enables a hierarchical and selective approach to noising different parts of the data. This flexibility allows the model to prioritize important features by applying noise schedules tailored to specific components, leading to a more efficient and accurate generative process. Combined with its support for multiple basis representations—such as pixel, PCA, Fourier, and wavelet bases—GUD is adaptable to various data types and structures, making it particularly suitable for applications that benefit from multi-scale or hierarchical data representations.

Additionally, GUD’s design includes a whitening process, which aligns the data and noise distributions, providing better variance control throughout the generative process. This feature simplifies denoising and increases model stability, potentially reducing training time by minimizing noise-related artifacts. By supporting flexible basis selection, component-wise noise control, and variance alignment, GUD allows for refined generative modeling that can adapt to diverse tasks and applications, offering a powerful tool for high-quality, customizable data generation.

**Weaknesses:**

The GUD framework is flexible, and consequently introduces significant computational complexity. Each configuration, such as basis choice (PCA, Fourier, wavelet) and component-wise noise scheduling, requires tuning, making the model resource-intensive. This complexity can hinder scalability, especially in high-dimensional data applications where each choice impacts the computational load.

Architecturally, GUD’s design adds complexity by requiring modifications like cross-attention mechanisms for conditioning on component-wise noise states. These additions complicate the implementation and increase the risk of instability during training, as standard architectures like U-Nets are not inherently optimized for GUD’s intricate conditioning needs. This limitation might however be not so crucial.

Finally, I think the authors could have expanded the comparison with  related works. As (non exhaustive) examples, non isotropic noise perturbation has been considered in [1] and optimal steady state covariance wrt the data distribution has been investigated [2].


[1] Voleti et al, Score-based Denoising Diffusion with Non-Isotropic Gaussian Noise Models, NeurIPS 2022 Workshop on Score-Based Methods.

[2] Das et al, Image generation with shortest path diffusion, ICML 2023

**Questions:**

Could the authors explain how to select in the large design space the various parameters/hyperparameters?

Can the authors briefly position wrt the works like the ones cited in the weaknesses section?

Minor:  Figure 7 is qualitatively difficult to interpret from someone not specialized in the field. I suggest the authors to either add some extra comments or produce a similar image for a dataset which is more understandable for a generic reader.

---

> ### Author Response · Authors · 2024-11-22
> **Design space, related work & further explanation**
>
> We thank the reviewer for the thoughtful questions and helpful suggestions.
>
> > Could the authors explain how to select in the large design space the various parameters/hyperparameters?
>
> We agree that this is an important question and one we are currently investigating further. As discussed in this paper and confirmed in the numerical experiments, we believe that the ordering of noising of the different components, and how sharply (autoregressively) we noise these has a significant impact. In this regard, it is important to stress that the optimal choice will depend on the dataset. To demonstrate the impact, in this work we experimented with choices informed by well-understood properties of natural images such as their power spectrum and multi-scale nature, including the PCA and Fourier basis and sequentially noising the respective components from small variance to large variance.
>
> > Can the authors briefly position wrt the works like the ones cited in the weaknesses section?
>
> We will discuss these two previous papers in the updated version. In short, the noising processes in these works can be regarded as special cases of the unifying framework we propose.
>
> > Minor: Figure 7 is qualitatively difficult to interpret from someone not specialized in the field. I suggest the authors to either add some extra comments or produce a similar image for a dataset which is more understandable for a generic reader.
>
> We will add an additional figure and more detailed explanations in the updated version to better explain the procedure.

---

> ### Author Response · Authors · 2024-11-26
> **Clarification and further comments on architecture design**
>
> >Architecturally, GUD’s design adds complexity by requiring modifications like cross-attention mechanisms for conditioning on component-wise noise states. These additions complicate the implementation and increase the risk of instability during training, as standard architectures like U-Nets are not inherently optimized for GUD’s intricate conditioning needs. This limitation might however be not so crucial.
> ---
> We thank the reviewer for the thoughtful comment on the architecture complexity. In what follows we clarify the following three points.
>
> - **noising state conditioned network**
> As GUD allows for choosing the SDE forward process that connects the data distribution (approximately) to the Gaussian prior by choosing a path for all components of the noising state vector $\gamma$ (as illustrated in Figure 1), it is indeed natural for us to consider a $\gamma$-conditioned network. In particular, conditioning on gamma instead of on time (for some given schedule) allows for one score network to (approximately) capture a range of different noising schedules, as we explain in the second paragraph of section 4.5. In the work we explored how an expressive $\gamma$-conditioned network may look like, in particular using cross attention. We have not encountered severe limitations in our experiments.
> In particular, we note that for the experiments in our paper it would in fact suffice to condition the network on just two or three scalar variables as opposed to the whole $\gamma$ vector, as in our experiments we limit ourselves to one or two dimensional families of schedules.
>
> - **for a given component-wise schedule**
> We would like to emphasize that, once a (component-wise) schedule is chosen, any standard score network conditioned on time can be used and $\gamma$-conditioning will not be necessary to run a GUD model with a fixed schedule.
>
> - **supporting numerical experiments**
> In part to demonstrate the above points, we have re-done the experiment of Figure 3 using a minimally modified version of Song et al's [2011.13456] NSCN++ architecture without cross attention. Having trained for more training steps than previously, we now obtain significantly better FID scores.

---

> ### Author Response · Authors · 2024-11-27
> **an updated version**
>
> We thank the reviewer again for the helpful comments and suggestions. We have uploaded a new version of the manuscript, in which we hope to have addressed the points you raised. Below are the main points.
> - We have expanded section 2 on related works and in particular commented on the work by Voleti et al and Das et al.
> - We have added a second panel (the right panel) to Figure 7 and expanded and edited the text around it to explain the procedure better.
> - In view of the comments on the architecture, we added a final paragraph to the Introduction and a final paragraph to section 4.5 (both in blue text), to clarify the points we wrote in the previous response.
>
> Please let us know if you have further questions or would like to see further improvements.

---

> > ### Comment · Reviewer_V72p · 2024-12-03
> >
> > I appreciate the authors' effort and the improvements made to the paper. After reviewing the rebuttal addressed to me and the other reviewers, I remain convinced that the paper meets the acceptance threshold. However, the concerns raised by other reviewers regarding the completeness of the experimental campaign prevent me from raising my score.

---

### Official Review · Reviewer_pLKS · 2024-11-04

**Soundness:** 3
**Presentation:** 3
**Contribution:** 2
**Rating:** 5
**Confidence:** 4

**Summary:**

The authors describe diffusion models with a under class of dynamics and marginals than the typical scaled Ornstein Uhlenbeck or Brownian motion reference processes used in the vast majority of diffusion model papers. In particular, the authors consider a linear transformation to perform the diffusion under a change of basis; varying the variance of the prior  Gaussian marginal  to match the data distribution; considering time dependent diffusion scale terms which can lead to auto-regressive-like dynamics.

**Strengths:**

The paper is well explained.

The authors bring attention to the flexibility in the diffusion model paradigm, though as discussed below this has been discussed in many prior papers.

The authors introduce what I believe to be a novel interpretation and use case for time-varying diffusion scale timers, leading to an autoregressive type forward process, applying noise to separate components independently. A similar procedure was used for diffusion in frequency space by applying different diffusion noise scales per frequency level [1] but these were not set to 0 as described here.


[1] Blurring diffusion models, Hoogeboom et al 2022

**Weaknesses:**

## Weakness 1
While the authors attempt to unify the design of dynamics for references; two of the three ideas proposed are not novel so it is unclear what the main contributions of the paper are.

1) Using a change of basis
Applying diffusion in a transformed space / change of basis has been done before. Although [1] focuses on  change of basis to frequency basis, section 4.1 of [1] explicitly explains how any other change of basis can be performed.  I do not see any compelling evidence to suggest one basis over another in this submission.

2) Prior distribution
Discussion of variance of prior distribution was first discussed in [3], and referred to as Technique 1. This is still using diagonal covariance.

It is not clear how scalable learning the covariance matrix for a full high dimensional data distribution would be or if it would even be beneficial.

The time-dependent diffusion scale term has not been investigated in much detail as far as I am aware and I believe this should be the main focus of the paper or at least more attention.

## Weakness 2
The second major weakness is in limited numerical evaluation. The FID scores shown for CIFAR10 are >20; significantly far from standard diffusion model performance of <3. It is not possible to evaluate whether there any benefit to generative modelling for the proposed methods without compelling numerical support.

Whilst I am not particularly interested in SOTA generative models FID <2, for toy datasets like CIFAR10 I would expect at least FID<4 given the abundance of code available for this and the limited novelty for 2/3 methods.

## Weakness 3
It is not clear to me the theoretical soundness of using the autoregressive approach for extending existing images i.e. changing dimension from previously trained model. It seems the generative process is no longer related to the time reversal of an SDE given the dimension changes. Can this be formalised?

Minor
- Blurring diffusion models [2] was a follow up to inverse Heat Dissipation Generative Model [1]. This should be cited and discussed as it was a pioneering paper in this area.


[1] GENERATIVE MODELLING WITH INVERSE HEAT DISSIPATION, Rissanen et al 2022
[2] Blurring diffusion models, Hoogeboom et al 2022
[3]Improved Techniques for Training Score-Based Generative Models, Song and Ermon, 2020

**Questions:**

See weaknesses.

The time-dependent diffusion scale term has not been investigated in much detail as far as I am aware and I believe this should be the main focus of the paper or at least more attention. What are the benefits of this compared to cascading diffusion, can cascading be seen as a case of this?

---

> ### Author Response · Authors · 2024-11-25
> **Clarifications and Experiments Results**
>
> We thank the referee for the thoughtful  comments. In what follows we will address the questions.
>
> ---
>
> On "Weakness 1":
> > It is not clear how scalable learning the covariance matrix for a full high dimensional data distribution would be or if it would even be beneficial.
>
> We would like to clarify that our work does not rest on learning the covariance matrix. In many cases this can be estimated directly from the dataset, and where this is not feasible one could use a power-law approximation to the power spectrum. More generally, our model does not rely on the noise scale exactly matching the data distribution, and such a choice may indeed not be the optimal one.
>
> >The time-dependent diffusion scale term has not been investigated in much detail as far as I am aware and I believe this should be the main focus of the paper or at least more attention.
>
> We agree that the component-wise schedule is the most important element of the GUD framework, and it has indeed been by far the main focus on our research effort. We will highlight this more clearly in the paper. The reason why we included the prior and the basis in the discussion of the design choices of the GUD framework is that they are inseparable and the schedule cannot be described without specifying the other two: the diagonal basis dictates what the "components" are in the component-wise schedule, and the prior dictates the endpoint of the distribution path that the component-wise schedule leads to. We do not see the inclusion of the other two choices as weakness.  We also thank the referee to remind us of the two papers which used a "non-standard" basis or prior. We will include a discussion on them in the manuscript.
>
> ---
>
> On "Weakness 2": We agree with the referee that the FID reported in Figure 3 is high. As mentioned in the footnote, we abruptly ran out of compute available to us. We have in the meantime resumed training with a somewhat modified network and achieved a much lower FID as a result. This part of the manuscript will be updated accordingly.
>
> ---
>
> On "Weakness 3":
> > It is not clear to me the theoretical soundness of using the autoregressive approach for extending existing images i.e. changing dimension from previously trained model. It seems the generative process is no longer related to the time reversal of an SDE given the dimension changes. Can this be formalised?
>
>
> One does not need to change the dimension from previously trained model in the application of image extension discussed in the paper. In particular, the generative process  still corresponds to the reverse time SDE, as we described in section C of the appendix. Instead, though the image gets larger in the process, the "active" dimension that participate in the generative process at any given diffusion time remains the same, while the already generated portion of the image remains frozen. In the updated manuscript, we will add additional figure and extra comments to explain this procedure better.
>
> In summary, the soft autoregressive forward process only noises a subset of columns at any time. The score network thus only has to output the score for this relevant region, outside of which the SDE is either frozen or exactly generates Gaussian noise. Assuming the distribution only depends on a finite region (neighboring columns), we can repeat the generative process by extending the image with noise and sliding the input to the score network over the generated image. More formally we can think of the whole large image initialized with white noise, across we then slide the score network to compute the non-trivial part of the SDE.
>
> ---
>
> Finally we thank the reviewer for pointing out reference [1]. We will include the citation in the manuscript in the part where relevant work is discussed.
>
>
> **In summary:**
>
> We believe that our main contributions are:
>
> 1. To explain and interpret the linear SDE ansatz in terms of the three degrees of freedom, unifying many specific choices discussed in previous works.
> 2. To give theoretical heuristic arguments for component-wise schedules in particular with regards to the discussion of soft autoregressiveness.
> 3. To give experimental evidence that such choices have a significant impact on performance.
>
> We will expand on the discussions on related literature including the 3 papers the referee cited, and edit the manuscript to highlight the component-wise schedule as the referee suggested. Finally, we will provide an updated version of experimental results.
>
> Please let us know if there is anything else we could further clarify or strengthen.

---

> ### Author Response · Authors · 2024-11-26
> **Cascading Diffusion Comparison**
>
> > The time-dependent diffusion scale term has not been investigated in much detail as far as I am aware and I believe this should be the main focus of the paper or at least more attention. What are the benefits of this compared to cascading diffusion, can cascading be seen as a case of this?
>
> We thank the reviewer for recognizing the novelty of this aspect of the our work and for giving us the opportunity to further explain the difference between our approach and cascading diffusion.
>
> In cascaded diffusion models, images are upscaled by training conditional diffusion models that generate higher resolution images with a low resolution image as input. The complete high-resolution image is generated from a noise sample. In our framework, one can simply make the corresponding choice for the basis and component-wise schedule to achieve a similar effect, albeit in a fundamental different way as we will now explain. If the components are chosen to represent coarse to fine grained linear subspaces of the image (e.g. Fourier or wavelet components), then the image can be generated from low to high resolution with any desired (soft) autoregressive ordering in our model. The coarse degrees of freedom are frozen after they have been fully denoised by virtue of the schedule, and are generally not provided as a separate conditional input. As a result, the coarse degrees of freedom in the generated high resolution image exactly match the low resolution image genearated earlier in the generation process in our model. This is to be constrasted with the cascaded diffusion model, in which the coarse degrees of freedom in the high resolution image only implicitly match the low resolution image that one conditions, through the training target. Finally, we note that coarse to fine is only one class of generative processes captured by our framework, and the continuous choice of softness (i.e. to what degree generation of degrees of coarseness overlap) is manifest in our framework but not in cascaded diffusion models. This example highlight the flexible and unifying nature of our framework.

---

> ### Author Response · Authors · 2024-11-27
> **updated version**
>
> We thank the reviewer again for the helpful comments and suggestions. We have uploaded a new version of the manuscript, which we hope to have improved based on the points you raised. Below are the main points.
>
> - On page 1, with the text in blue we highlight that the component-wise schedule is the most important element of the GUD framework.
> -  We have expanded the section 2 on related work. We have in particular expanded the discussions on blurring diffusion, and included references to [Generative Modelling with Inverse Heat Dissipation] and [Improved Techniques for Training Score-Based Generative Models].
> - We have updated Figure 3 (page 8) to reflect the lower FID after further training. In relation to the comments on available codes, we also amended section 4.5 with text in blue to report on further experimentations in this direction. To clarify the nature and the main purpose of our numerical experiments, we added a small paragraph at the end of the introduction. In particular, we expect that given a fixed schedule one can use a simpler network and fine-tune it to improve the resulting performance. However, this would not serve the primary purpose of the experiments of this paper.
>
> - Given the misunderstanding ("weakness 3") of the image extension experiments, we have further explained what we did with edited text in blue and by adding a new Figure (the right panel of Figure 7), on p.9-10.
>
> Please let us know if you have further questions or would like to see further improvements.

---

> > ### Comment · Reviewer_pLKS · 2024-11-28
> >
> > Thank you for taking the time to respond; including missing references and clarifying prior works which introduced change of basis for diffusion models.
> >
> > Figure 3 still show scores 6-7; which is significantly worse than basic diffusion models from 3-4 years ago or so; and extremely far away from recent results. It is really not clear how to evaluate whether the improvements are useful with such results. I sympathise with lack of compute but CIFAR10 is the smallest and simplest dataset that could be considered.
> >
> > Even for the column-wise extension, it appears only a single dataset has been used?
> >
> > I find it very difficult to accept such a paper without proper numerical evidence of being useful.

---

> ### Author Response · Authors · 2024-11-28
> **focus of the experiments**
>
> We thank the reviewer for sharing the thoughts on this. Just for clarification 1. in figure 3, the most optimal point has FID 5.26.  2. We showed PCAM dataset for column-wise extension. 3. We added an appendix on celebA experiments, again focussing on the comparison between schedules and not on the absolute performance.
>
> We would like to emphasize again that we consider our main contribution to be to offer a unifying framework in which many optimizations can take place that  aree bound to lead to improvement, given a specific task. In this spirit, our experiments focus on the **comparative study** between different choices: to **show that the choices make a difference in performance**, and that **the standard diffusion does not seem to be optimal (which is hardly surprising)**, and **to corroborate the advantage of incorporating more autoregressiveness.** Given choices, hyperparameters than need to be tuned individually to achieve better perforrmances.
>
> We hope this clarifies why we think our paper makes valuable contribution to the study of diffusion generative models.

---

### Meta-Review · Area_Chair_CXew · 2024-12-20

**Metareview:**

This paper proposes the Generative Unified Diffusion (GUD) framework, which unifies diffusion and autoregressive models by introducing flexibility in data representation, noise scheduling, and prior distributions. While the theoretical foundation is intriguing and contributes to the design space for diffusion-based generative models, the submission lacks sufficient empirical evidence to validate its practical utility.

**Additional Comments On Reviewer Discussion:**

Several reviewers highlighted the strength of the GUD theory, particularly its use of component-wise noise scheduling. However, they also raised issues such as limited experimental scope, incomplete evaluations on larger datasets (e.g., CelebA-HQ), and insufficient comparisons with related works. Additionally, the paper fails to provide convincing ablations or empirical support to isolate and quantify the benefits of its contributions. While the proposed ideas have potential, the current version does not demonstrate sufficient rigor or results to justify acceptance.

---

### Decision · Program_Chairs · 2025-01-22

Reject